# RL-Guided Data Selection for Language Model Finetuning

## Abstract

Data selection for finetuning Large Language Models (LLMs) can be framed as a budget-constrained optimization problem: maximizing a model's downstream performance under a strict training data budget. Solving this problem is generally intractable, and existing approximate approaches are pretraining-oriented and transfer poorly to the fine-tuning setting. We reformulate this problem as a tractable Markov Decision Process (MDP) and train agents using various Reinforcement Learning (RL) methods to learn optimal data selection policies, guided by an efficient, proxy-model-based reward signal. Across four datasets, training on a $5\%$ subset selected by our approach matches or outperforms fine-tuning on the full dataset by up to $10.8$ accuracy points, while cutting wall-clock training time by up to $2\times$, highlighting the promise of RL-guided data selection.

## 1  Introduction

Real-world datasets for LLM finetuning often contain noisy and redundant data points [8], which inflates computational costs and can degrade model performance [12]. Strategic data selection methods offer a solution by identifying a small, high-quality training subset [24, 23]. These methods solve a budget-constrained combinatorial optimization problem: maximize a model's downstream performance while adhering to a strict data budget, typically a fixed fraction of the original dataset.

Provably solving this optimization problem is intractable due to the exponential search space and prohibitive evaluation costs. While performant and approximate data selection methods have been developed for large-scale pre-training [24, 23], they are ill-suited to the finetuning regime. They are often prohibitively expensive for the smaller scales typical of finetuning datasets [23] and largely capture surface-level patterns rather than task-specific semantics [7].

To bridge this gap, we introduce a framework that reformulates the problem of data selection as a tractable Markov Decision Process (MDP). We first group the training data into semantic clusters, defining a state space over subsets of these clusters. Actions are defined as sequentially adding new clusters to the training subset corresponding to the current state. An RL agent then learns a selection policy, guided by an efficient proxy of the downstream performance objective, derived from a smaller model's validation loss on selected data subsets.

Across four diverse tasks [9, 19, 17], training on a $5\%$ subset selected by our approach matches or even significantly exceeds the performance of training on the full dataset and other heuristic baselines, while also cutting wall-clock times by up to $2\times$. Notably, on MetaHate [17], our approach boosts accuracy by $10.8$ points over the full-data baseline, showing that it can filter out harmful, noisy and unreliable data. We conclude that RL-guided approaches achieve a good balance between downstream performance and training efficiency, demonstrating substantial potential for data subset selection in LLM fine-tuning.

Submitted to the NeurIPS 2025 Workshop on Reliable ML from Unreliable Data. Do not distribute.

## 2 Related Work

The goal of data selection is to identify a subset of training data that preserves downstream performance while adhering to a data budget. In 11, a statistical theory is proposed for data subsampling under weak supervision across a variety of model classes. This is extended to frame data selection as an information-theoretic problem in 4. On the other hand, DSDM [6] and Influence Distillation [18] introduce model-aware approaches to analyze the influence of individual data points on specific target samples. Finally, 7 reformulates data selection as an optimal control problem solvable via Pontryagin's Maximum Principle. In contrast, this work formalizes data selection as a budget-constrained combinatorial optimization problem, which is reduced to a tractable Markov Decision Process.

Data selection for LLM training has also been extensively studied in recent literature, given the ever-growing scales of training datasets [2]. The LESS framework [23] quantifies the contributions of individual samples to model convergence by constructing gradient stores, but has high computational cost [26, 14]. In contrast, methods such as DSIR [24] utilize importance resampling to select examples that are statistically most beneficial for pre-training, while DoReMi [25] optimizes data mixtures to accelerate language model pretraining. Other strategies include data pruning [15] and deduplication methods like D4 [22] and SemDeDup [1] that aim to improve training efficiency by reducing redundancy. More recently, CLIMB [5] iteratively samples random data mixtures, evaluates them, and trains a predictor that guides subsequent mixture selection. RL has remained largely unexplored in the context of LLM fine-tuning in contemporary literature.

## 3 Methodology

### 3.1 Data Selection as a Constrained Optimization Problem

Given a training dataset $D$, we seek to identify a subset $S \subseteq D$ that minimizes the test loss of a model $M$ trained on $S$ as computed on a held-out test set $D_{test}$, subject to a cardinality constraint $|S| \leq K$. This can be formulated as the following optimization problem:

$$S^* = \arg \min_{S \subseteq D, |S| \leq K} \mathcal{L}_M(S|D_{test}) \tag{1}$$

where $\mathcal{L}_M(S|D_{test})$ is the loss obtained on $D_{test}$ when $M$ is trained on $S$. Solving this problem is intractable, since the objective function is non-differentiable with respect to $S$, and evaluation for any $S$ requires model training on $S$. Therefore, we approximate the solution set $S^*$ as the solution to a tractable sequential MDP, described in the next section.

### 3.2 A Tractable MDP Formulation

We first cluster the training dataset $D$ into a set of semantically coherent clusters $C$ via K-Means clustering on sentence embeddings (more details in Appendix A). The MDP is then defined over the powerset of these clusters, $\mathcal{S} = \mathcal{P}(C)$. A state $s_t \subseteq C$ represents the subset of clusters selected up to time step $t$. From a state $s_t$, the agent can select any cluster not already in the current subset ($\mathcal{A}_{s_t} = C \setminus s_t$). Transitions are deterministic, with $s_{t+1} = s_t \cup \{a_t\}$. Each episode proceeds for a fixed horizon H, terminating when the subset size $|s_H|$ reaches the budget defined by the selection fraction $\delta|C|$. Each episode of the MDP corresponds to the sequential selection of a set of clusters to form a possible training data subset, while adhering to the data budget enforced by $\delta$. This MDP is tractable for small $|C|$. We study the effect of varying $|C|$ in Appendix E.

### 3.3 Reward Function

We define the reward function $R(s_t, a_t)$ for the MDP as the change in validation loss from a proxy model $M'$ when the cluster $C_t$ (selected during the action $a_t$) is added to the training data subset represented by the state $s_t$. $M'$ is typically a smaller model in the same model family as the target model $M$. To improve the efficiency of reward computation, we further subsample the data points in each cluster belonging to $C$ using a subsampling function $\xi(\cdot)$. Formally:

$$R(s_t, a_t) = f(\mathcal{L}_{M'}(\xi(s_t) \cup \xi(\{a_t\})|\xi(D_{\text{val}}))) - f(\mathcal{L}_{M'}(\xi(s_t)|\xi(D_{\text{val}}))) \tag{2}$$

where $\mathcal{L}_{M'}(D_t|D_v)$ is the loss on validation set $D_v$ after training $M'$ on training set $D_t$, and $f(\cdot)$ is a logarithmic transformation to amplify small loss variations. More details can be found in Appendix A. This reward signal serves as a computationally efficient proxy for the downstream performance.

## 3.4 Learning a Sequential Data Selection Policy

We leverage our MDP formulation to learn a policy $\pi(s_t)$ for selecting the next cluster to add to the current subset $s_t$. The final data subset is then constructed by starting with an empty set and iteratively applying the learned policy for a predefined number of steps corresponding to the desired selection fraction. We try several RL algorithms to learn the policy, including Deep Q-Networks (`DQN`) [16] and Proximal Policy Optimization (`PPO`) [20]. For PPO, we also tried a `Warm-Start` initialization by pre-training the critic model on a regression task over the rewards of single-cluster states. However, a naive exploration of the state space is intractable due to its exponential size $(2^{|C|})$. To mitigate this, we augment the reward function with a bonus derived from Random Network Distillation (`RND`) [3], which incentivizes the policy to visit novel state configurations.

The computational cost of reward evaluation remains a bottleneck even with a proxy model. Therefore, we investigate model-based strategies for learning an explicit, lightweight reward function to be used for generating synthetic rollouts. Our first approach (`DynaDQN`) is inspired by Dyna [21] and integrates a learned reward model with `DQN`. The reward model is used to label synthetically generated state-action pairs, which are then added to the replay buffer to accelerate learning. Our second approach (`CLIMB-Disc`) is an adaptation of CLIMB [5] with discrete cluster selection. Specifically, it is a form of Bayesian search, where the trained reward model is used as a sampling prior. At each step, we sample a batch of unseen states, use the model to identify the top candidates, query their true rewards to update the model, and repeat. Further details are provided in Appendix B.

# 4 Experiments

## 4.1 Experimental Setup

**Datasets:** We use the MMLU [9], ANLI [17], MetaHate [19] and GooglePlay datasets (more details in Appendix D). The MetaHate and GooglePlay datasets do not have an explicit test split, so we randomly sample 25K and 5K samples respectively to create one. We fix the data selection percentage to 5% of the full training dataset unless otherwise mentioned.

**Models:** MobileLLM-600M [13] serves as the proxy model for reward computation, and MobileLLM-1.5B is used as the target model for final evaluation.

**Baselines:** We compare against training the target model on (a) `Full`, the entire training dataset; (b) `Random`, a randomly selected 5% of the training dataset; (c) `Top-Loss`, the 5% of the dataset with the highest loss as computed by the proxy model; (d) `Bottom-Loss`, the 5% of the dataset with the lowest loss as computed by the proxy model; (e) `Random-Search`, performing random rollouts from our MDP, scoring them using our reward function, and selecting the rollout with the highest reward. We provide hyperparameters for our experiments in Appendix C.

**Evaluation:** We report accuracy on a held-out test set for each dataset, for a target model trained on the data subsets selected by the different approaches.

## 4.2 Results

We present results for all approaches in Table 1. We find that RL-guided data selection significantly outperforms standard baselines across all tasks. In some cases, it even surpases the performance obtained by training on the full dataset, notably by 10.8 points for MetaHate and 0.3 points for GooglePlay. We conclude that our learned policies mitigate the deleterious effects of noisy data points for these datasets, by filtering them out. All RL policies also consistently outperform all random selection and heuristic baselines.

We find that the `Random-Search` baseline improves upon `Random`, validating that our reward is a meaningful proxy for downstream performance. The superior performance of DQN, PPO and

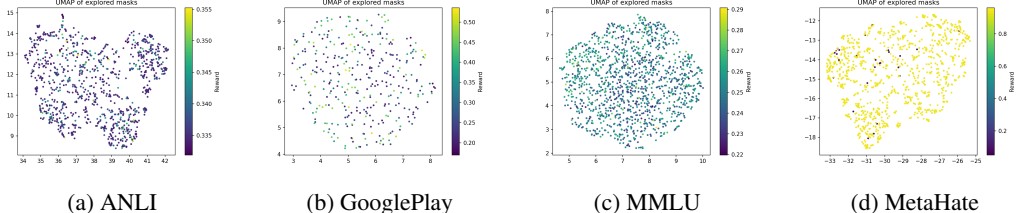

| | (a) ANLI | (b) GooglePlay | (c) MMLU | (d) MetaHate |

Figure 1: UMAP projections of explored state (binary mask) encodings, colored by their subsampled validation set accuracy.

| Algorithm | ANLI | GooglePlay | MetaHate | MMLU |
|---|---|---|---|---|
| Full | 64.76 | 68.10 | 83.20 | 49.38 |
| Random | 54.20 | 58.30 | 72.60 | 40.90 |
| Top-Loss | 57.40 | 21.90 | 84.00 | 37.34 |
| Bottom-Loss | 57.10 | 22.60 | 77.80 | 22.96 |
| Random-Search | 55.61 | 59.30 | 72.60 | 43.71 |
| DQN | **57.60** | 65.60 | 69.40 | 44.27 |
| DQN + RND | 35.30 | 63.76 | 70.91 | 44.18 |
| PPO | 54.20 | 62.32 | 60.85 | 44.80 |
| PPO + Warm-Start | 56.24 | 60.24 | 87.95 | 44.19 |
| PPO + RND | 55.80 | 56.52 | 59.50 | **45.68** |
| DynaDQN | 52.96 | 61.94 | 50.50 | 45.11 |
| CLIMB-Disc | 53.83 | **68.40** | **94.01** | 41.73 |

Table 1: Performance of MobileLLM-1.5B when trained using the different data selection strategies discussed in Section 3.4. The best numbers across the approaches are highlighted.

CLIMB-Disc over Random-Search further indicates that these approaches learn meaningful, nuanced selection policies. However, we note that the best approach changes for each dataset. While the Warm-Start initialization for PPO improves performance on ANLI and MetaHate by up to 27.1 points, the RND bonus did not yield meaningful benefits.

We hypothesize that the comparative success of our method on MetaHate and GooglePlay is linked to the diversity of their reward landscape. As visualized in Figure 1, these datasets exhibit high reward variance across different clusters. In contrast, ANLI, which has the lowest reward variance, shows the largest remaining gap to the full-data baseline. This suggests that our MDP formulation is particularly potent for noisy datasets where the value of intelligent data selection is highest.

Finally, our method offers a compelling trade-off between performance and efficiency. By training on a curated 5% of the training data, we achieve strong results in less than half the wall-clock time of full-dataset training, including the overhead of the data selection process (but excluding the overhead of hyperparameter search). Detailed results and ablations are provided in Appendix E.

## 5 Conclusion

We propose a RL-based framework for solving the budget-constrained optimization problem of data selection for LLM fine-tuning. We reformulate the task as the solving of a tractable MDP over clusters of the training data, and train RL agents to learn policies for sequentially constructing high-quality data subsets using an efficient proxy-based reward. We find that our approach is effective in practice across four diverse datasets. In fact, training on a 5% data subset selected using our approach often exceeds the performance obtained by training on the full dataset by filtering out unreliable, noisy or redundant data points, with significant training efficiency gains. We conclude that RL-based approaches are effective for approximately solving this important constrained optimization problem.

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

## A Detailed Methodology

Here we provide additional details on the MDP formulation, state representations, reward functions, and policy learning algorithms explored in this work.

### A.1 Clustering

In addition to standard K-means clustering we also try to induce label information in the clusters, for this we tried a variant where we enforce a cluster to have data points corresponding to only one label (henceforth called Stratified-Kmeans)

### A.2 State Representations and Subsampling

For a given state $s_t$, we explore different ways of computing a state encoding $\phi(s_t)$. The simplest encoding, denoted by Binary-Mask, is $|C|$-length binary vector with $\phi_i(s_t) = 1 \Leftrightarrow C_i \in s_t$. In another case (Mean-Std), we use:

$$\phi(s_t) = [\mu(s_t), \sigma^2(s_t)],$$

where $\mu(\cdot)$ and $\sigma^2(\cdot)$ are the mean and variance of the cluster-centroid embeddings in the currently selected set. Another variant (Concat) involves concatenating embeddings of representative samples from each cluster. We explore two approaches for selecting these representative samples choosing them at random from the cluster (Random) or choosing the furthest points from the cluster centroid (Furthest), capturing the spread of the cluster.

### A.3 Reward Functions

In our experiments, we evaluated three distinct reward functions. All are computed using the proxy model, $M'$, which is a smaller version of the target model, $M$. The primary reward signal as detailed in the main section is $R_{\mathbf{loss}}^{\mathbf{val}}$ which is based on change in validation loss. Let Val-Acc$(D)$ be the accuracy of the proxy model $M'$ on the validation set after training on dataset $D$ and $\mathcal{L}_{M'}(\mathbf{D_v}|\mathbf{D_t})$ be the loss value for dataset $\mathbf{D_v}$ after training $M'$ on $\mathbf{D_t}$. (for clearness, we omit $\mathbf{D_t}$ if it is same as $\mathbf{D_v}$, we also omit $M'$ as all rewards are computed using the proxy model)

**Accuracy-based Reward** ($R_{\mathbf{acc}}$): This reward function computes the improvement in validation accuracy when adding a new cluster to the selected data, thus capturing its impact on the downstream performance of the proxy model:

$$R_{\text{acc}}(s_t, a_t) = \text{Val-Acc}(s_t \cup \{a_t\}) \; - \; \text{Val-Acc}(s_t). \tag{3}$$

Although effective, measuring changes in validation accuracy entails retraining the proxy model from scratch after each action for a substantial number of training steps and performing evaluation, which is extremely expensive.

**Training Loss-based Reward** ($R_{\mathbf{loss}}^{\mathbf{train}}$): This reward function makes two assumptions — training losses on the same batches of data are correlated for the *target* and *proxy* model, and training loss for a model is negatively correlated with downstream performance. Then, the reward function measures changes in the proxy model's training loss when the new cluster is added to the current state:

$$f(x) = 5 - 2\ln(2x) \tag{4}$$
$$R_{\text{loss}}(s_t, a_t) = f\big(\mathcal{L}(\xi(s_t) \cup \xi(\{a_t\}))\big) \; - \; f\big(\mathcal{L}(\xi(s_t))\big). \tag{5}$$

where $\ln(\cdot)$ is the natural logarithm, and a subsampling function $\xi(\cdot)$ is used to select a fixed number of data points (set as a hyperparameter) from each cluster to estimate the training loss from the *proxy* model at the end of multiple epochs of training. The logarithmic transformation $f(\cdot)$ serves a dual purpose: it establishes a baseline of $f(\mathcal{L}(\emptyset)) = 0$ while also magnifying subtle loss variations in the low-loss regime of training on larger subsets of data. $R_{\mathbf{loss}}$ is much faster than $R_{\mathbf{acc}}$, which makes MDP rollouts more efficient.

**Validation Loss-based Reward ($R_{\text{loss}}^{\textbf{val}}$):** This reward function is similar to $R_{\text{loss}}^{\text{train}}$, except for using validation-set loss instead of training loss. Formally,

$$R_{\text{loss}}^{\text{val}}(s_t, a_t) = f\big(\mathcal{L}(\xi_{val}(\mathbf{D}_{val})|\xi(s_t) \cup \xi(\{a_t\}))\big) - f\big(\mathcal{L}(\xi_{val}(\mathbf{D}_{val})|\xi(s_t))\big). \tag{6}$$

where the subsampling function $\xi_{val}(\cdot)$ is used to select a fixed number of data points (set as a hyper-parameter) from the validation set, keeping the label proportion constant. $f$ serves a similar purpose to that in $R_{\textbf{loss}}$. While $R_{\textbf{loss}}^{\textbf{val}}$ is slower than $R_{\textbf{loss}}^{\textbf{train}}$), it is much better correlated with downstream performance.

**Random Network Distillation (RND):** For each of the reward approximations described above, Random Network Distillation [3] can be added to improve exploration of the policy. RND is implemented using a 4-layer MLP with MSE loss between the target and predictor network as intrinsic reward. The state and rewards are normalized using a running average to stabilize the intrinsic rewards.

# B  Policy Learning Algorithms

`DQN`: At each state $s_t$, we compute an embedding $\phi(s_t)$ using one of the state encoding methods. We then feed $\phi(s_t)$ into a function approximator $f_\theta(\cdot)$, either an `MLP` or a small `Transformer`, which outputs an $|\mathcal{A}|$−dimensional vector where each component represents the estimated Q-value (or "goodness") of taking action $a \in \mathcal{A}$ in the current state $s_t$. We then mask out actions corresponding to the clusters already in $s_t$ and choose the action with the highest Q-value via $\epsilon$-greedy sampling. The network parameters $\theta$ are then optimized through experience replay updates.

`PPO`: We adopt a variant of PPO that supports the masking of invalid actions [10]. Both the actor and critic networks are 3-layer MLPs; for each state $s_t$, the actor outputs a probability distribution over available cluster actions, while the critic estimates the value of $s_t$. We investigate two variants of PPO as well. We first try training PPO from `Scratch`, initializing the actor and critic randomly. Next, we try to give PPO a `Warm Start`. We pre-train the critic using a regression task on rewards for "single-cluster" states. Specifically, for each cluster $c_i \in \mathcal{A}$, we compute the average reward obtained when taking action $c_i$ on the state containing the empty set to reach state $s_i$. We then regress the critic network on the $(s_0, c_i, s_i, r_i)$ tuples, where $s_0 = \emptyset$ and $r_i$ corresponds to the average reward for each single-cluster addition. This setup encourages the critic to produce, for the start state, outputs that rank clusters in proportion to their individual expected returns.

**Reward Model Based Strategies**

These strategies approximate the true reward function in order to accelerate policy learning by generating additional, "synthetic" rollouts. Concretely, we train a proxy reward model $\hat{r}_\phi(s, a)$ on true reward signals $r(s, a)$ and then use $\hat{r}_\phi$ to label transitions sampled under the current policy, and mix these synthetic transitions with real ones when updating the agent. Real rollouts are given higher weight. Based on the agent, we have two strategies: `DynaDQN` and `CLIMB-Disc`.

`DynaDQN`: The proxy reward $\hat{r}_\phi$ is implemented as an ensemble of four independently initialized, 5-layer MLPs. Each ensemble member is trained on real transitions using mean-squared error (MSE) loss with $\ell_2$ regularization. MLP variant of DQN is used as the policy. At each environment step, we sample a batch of 32 state–action pairs, compute their proxy rewards by averaging the ensemble outputs, and then only insert those synthetic transitions into the replay buffer if the ensemble standard deviation falls below a fixed threshold $\sigma_{\max}$. Synthetic transitions are retained for at most four episodes, and during learning, they are weighted by an importance factor of $0.5$ relative to real transitions.

`CLIMB-Disc`: Drawing inspiration from [5], we implemented `CLIMB-Disc` for discrete states. For this strategy, the reward function $r(s)$ is the absolute value instead of the increment from the previous state. The proxy reward model $\hat{r}_\phi(s)$ is a single 3-layer MLP trained with MSE loss. In each iteration, we uniformly sample $M$ previously unseen states, rank them by their estimated reward $\hat{r}_\phi$, then query the environment for the true reward of the top-$K$ states and use these $K$ new labels to update $\hat{r}_\phi$. After $T$ epochs, we re-evaluate all seen states under $\hat{r}_\phi$ and select the highest-scoring one as the final best state.

## C  Hyperparameters and Experimental Settings

`BAAI/bge-small-en-v1.5` is used to obtain semantic embeddings for the training datasets and K-Means or stratified K-Means clustering is used to cluster the resulting embeddings into 64 (or 128) clusters. We use a batch size of 16 with 4 gradient accumulation steps to train the proxy model for 2 epochs with a learning rate of 1e-5. For each cluster, 64 data points are sampled for proxy-model training.

For the DQN, we use a 5-layer `MLP` of size 256 to learn the Q-function, with `Mean-Std` state encodings and `Furthest` subsampling. We use $\gamma = 0.99$ and decaying $\epsilon$ starting from 1 with a decay of 0.99 per episode and a minimum of 0.01. A replay buffer is used and steps are sampled in batches of 32 to train the model. A learning rate of $10^{-4}$ is used to train the DQN network and the target network is updated every 10 steps. The DQN is trained for 500 episodes. PPO is trained with a learning rate of $3 \cdot 10^{-4}$, for 500 episodes. For the linear bandits approach, we train for 1000 steps with a UCB coefficient of 2 and learning rate of $10^{-4}$. In DynaDQN, the reward model has a hidden dimension of 256, and the same configuration as DQN is used for policy. Learning rate of $5 \cdot 10^{-4}$ is used with no training for first 5 episodes. CLIMB-Disc is trained with 50 iterations, sampling 128 states and selecting top 32 states finally at each step. The hidden dimension is set to 128, and learning rate of $10^{-4}$ is used with the reward model trained for 2 epochs per iteration.

We train the target model for 4 epochs on the selected data subsets, with a batch size of 4 and 8 gradient accumulation steps, and use a cosine annealing schedule for the learning rate from 1e-5 to 1e-6 and linear warmup for the first 5% of training steps. Checkpoints are chosen based on highest validation accuracy for all settings to compute downstream performance.

## D  Tasks

| Dataset | Task | Train Size | Test Size | # Labels |
|---------|------|------------|-----------|----------|
| ANLI | Natural Language Inference | 162,400 | 3,200 | 3 |
| MetaHate | Hate Speech Detection | 1,051,165 | 25,000 | 2 |
| GooglePlay | Sentiment Classification | 98,836 | 5,000 | 5 |
| MMLU | MCQ Answering | 99,842 | 14042 | 4 |

Table 2: Summary of datasets used in our experiments with their respective tasks, training sizes, test sizes, and number of labels.

## E  Additional Results and Ablations

### E.1  Number of Clusters

We evaluate `Random-Search` algorithm over a range of cluster counts $C \in \{64, 256, 1024, 4096\}$, with results shown in Figure 2. As $C$ increases, we observe a consistent improvement in the downstream performance. However, the total runtime grows approximately quadratically in $C$, since both the number of episodes and the number of proxy sub-samples per reward evaluation increase with the cluster count. Balancing this trade-off between solution quality and computational cost, we fix $C = 64$ and proxy subsamples to 64.

### E.2  Clustering Strategy

The `Stratified-Kmeans` method exhibits suboptimal performance when the number of clusters is small and the number of class labels is large. This is primarily due to its inability to ensure representation of all labels in the selected subset, which leads to label imbalance. However, as the number of clusters increases, its performance improves, as shown in Figure 2. This improvement is attributed to the greater flexibility in selecting samples with more diverse label distributions across an increased number of clusters.

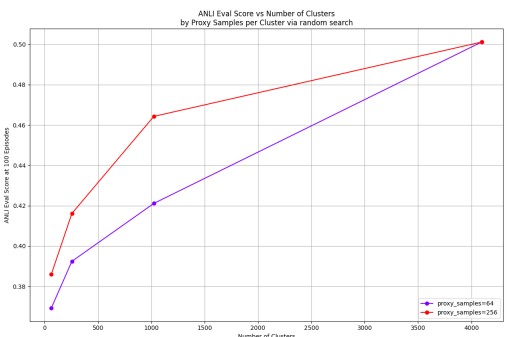

Figure 2: Downstream performance vs. number of clusters for ANLI with `Random-Search` and stratified k-means.

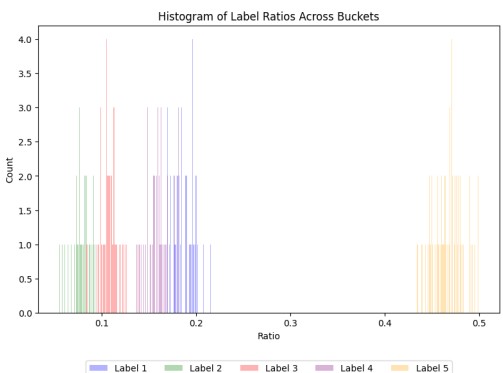

Figure 3: Histogram of label ratios across clusters using `K-means` in the GooglePlay dataset.

In contrast, `K-means` tends to preserve the overall label distribution more consistently, making it more effective when the number of clusters is limited. This distinction is illustrated in Figure 3, which presents the distribution of label proportions across clusters for the GooglePlay dataset. The figure demonstrates how label representation varies between the two methods and supports the superior performance of K-means in scenarios with fewer clusters.

## E.3 Comparison for Different State Encoders

| Dataset | State Representation | Subsampling | DQN Model | 600M Proxy Accuracy (↑) | 125M Proxy Accuracy (↑) |
|---|---|---|---|---|---|
| ANLI | `Mean-Std` | `Furthest` | `MLP` | **57.6** | **57.2** |
| | `Mean-Std` | `Random` | `MLP` | 52.9 | 54.6 |
| | `Concat` | `Furthest` | `Transformer` | 56.0 | 56.6 |
| | `Concat` | `Random` | `Transformer` | 54.9 | 53.2 |
| MetaHate | `Mean-Std` | `Furthest` | `MLP` | **69.4** | 63.4 |
| | `Mean-Std` | `Random` | `MLP` | 67.0 | 36.0 |
| | `Concat` | `Furthest` | `Transformer` | 60.9 | 61.6 |
| | `Concat` | `Random` | `Transformer` | 67.4 | **66.0** |
| GooglePlay | `Mean-Std` | `Furthest` | `MLP` | **65.6** | 60.6 |
| | `Mean-Std` | `Random` | `MLP` | 65.1 | 62.3 |
| | `Concat` | `Furthest` | `Transformer` | 61.8 | 59.4 |
| | `Concat` | `Random` | `Transformer` | 63.3 | **64.9** |

Table 3: Performance of MobileLLM-1.5B when trained on data selected using various DQN variants and two different proxy models. All strategies are discussed in Section 3.4. The best numbers for the data selection approaches are highlighted.

`DQN` : We present results for DQN methods with various state encoding methods, subsampling strategies, and DQN models across three datasets and two proxy models in Table 3. Our findings indicate that the `Furthest` subsampling strategy outperforms the `Random` strategy in nearly all cases, except for the 125M proxy model on GooglePlay and the `Transformer`-based DQNs on MetaHate and GooglePlay. Notably the additional expressive power provided by the `Transformer` does not generally lead to better performance compared to the `MLP`-based approach, except for the 125M proxy model on MetaHate and GooglePlay. Overall, using the 600M proxy model tends to yield better results for DQN-based approaches across all datasets. While there are no clear winners, using the `Mean-Std` state encoding with `Furthest` sampling and a `MLP`-based DQN results in generally strong performance across datasets.

`CLIMB-Disc` : We present the results for running CLIMB-Disc for multiple configurations of environments with `Furthest` subsampling in Table 4. Note that `Stratified-Kmeans` is run with 128/32 to allow for representation of all (5) labels in chosen clusters. From the numbers, we find that $R_{acc}$ with `Binary-Mask` performs the best in all configurations and 600M performs better than 125M.

| Clustering Type | # clusters/subsamples | Proxy Model | State encoder | $R_{\mathbf{acc}}$ | $R_{\mathbf{loss}}^{\mathbf{train}}$ | $R_{\mathbf{loss}}^{\mathbf{val}}$ |
|---|---|---|---|---|---|---|
| Kmeans | 64/64 | 125M | `Binary-Mask` | **65.50%** | 62.12% | 65.36% |
| | | | `Mean-Std` | 65.04% | 64.40% | 61.88% |
| Kmeans | 64/64 | 600M | `Binary-Mask` | **68.40%** | 64.40% | 65.58 |
| | | | `Mean-Std` | 62.62% | 63.84% | 59.42% |
| Stratified Kmeans | 128/32 | 125M | `Binary-Mask` | **61.38%** | 46.90% | 46.16% |
| | | | `Mean-Std` | 55.36% | 56.28% | 46.76% |

Table 4: Performance of MobileLLM-1.5B for GooglePlay dataset when trained on $1/16$ data selected using CLIMB-Disc with different state encodings and different reward functions.

| Dataset | Variant | 600M Proxy Accuracy (↑) | 125M Proxy Accuracy (↑) |
|---|---|---|---|
| ANLI | `Scratch` | 54.2 | 53.7 |
| | `Warm Start` | **55.8** | **54.9** |
| MetaHate | `Scratch` | 60.9 | 45.9 |
| | `Warm Start` | **73.1** | **88.0** |
| GooglePlay | `Scratch` | **62.3** | **61.7** |
| | `Warm Start` | 55.8 | 60.2 |
| MMLU | `Scratch` | **44.8** | - |
| | `Warm Start` | 44.19 | - |

Table 5: Performance of MobileLLM-1.5B when trained on data selected using PPO with and without warm starts and two different proxy models. The best numbers are highlighted.

Also, $R_{\mathbf{loss}}^{\mathbf{train}}$ performs better with `Mean-Std`, while $R_{\mathbf{loss}}^{\mathbf{val}}$ performs better with `Binary-Mask`. These results suggest that the semantic information presented in state by `Mean-Std` is not meaningful in case of validation set based rewards. Given the much higher time taken by $R_{\mathbf{acc}}$, $R_{\mathbf{loss}}^{\mathbf{val}}$ with `Binary-Mask` is the most suitable choice.

## E.4 Strategy Specific Comparisons

**PPO Warm Start** We present results for `PPO` with and without the `Warm Start` in Table 5 for all four datasets and two proxy models. The `Warm Start` is beneficial to the performance of PPO for both ANLI and MetaHate, but worsens performance slightly on GooglePlay and MMLU. Notably, the `Warm Start` nearly doubles downstream performance for MetaHate with the 125M proxy model.

**RND:** We evaluate the performance of the `RND` environment using $R_{\mathbf{loss}}^{\mathbf{val}}$ as the base reward signal with `DQN-MLP` and `PPO` policies. The corresponding results are presented in Table 6. It indicates that RND yields only marginal improvements in performance for the MetaHate task with `DQN-MLP` and the MMLU task with `PPO`, while substantially degrading performance across all other task–algorithm combinations. These results suggest that RND does not provide meaningful benefits for this MDP.

| Dataset | Variant | DQN Accuracy (↑) | PPO Accuracy (↑) |
|---|---|---|---|
| ANLI | `Val-Loss` | **57.6** | **56.24** |
| | `RND` | 35.3 | 55.8 |
| MetaHate | `Val-Loss` | 69.4 | **87.95** |
| | `RND` | **70.91** | 59.5 |
| GooglePlay | `Val-Loss` | **65.6** | **60.24** |
| | `RND` | 63.76 | 56.52 |
| MMLU | `Val-Loss` | **44.27** | 44.19 |
| | `RND` | 44.18 | **45.68** |

Table 6: Performance of MobileLLM-1.5B when trained on data selected using PPO and DQN with and without RND exploration reward. The best numbers are highlighted.

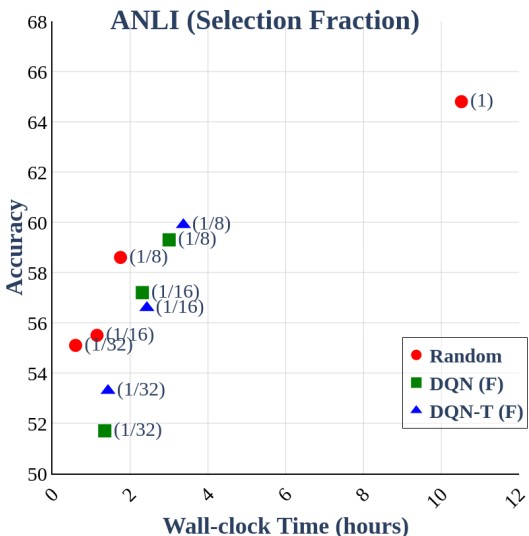

Figure 4: Downstream Performance vs Training Times for the Random and Full baselines, along with two DQN-based approaches.

**Reward Model Based Strategies:** Comparing the performance of various reward model based strategies in table 1, we find that `CLIMB-Disc` demonstrates consistently strong performance, out-performing all other strategies for GooglePlay and MetaHate. In contrast, while `DynaDQN` slightly surpasses `DQN` on MMLU, it underperforms significantly on ANLI, GooglePlay, and MetaHate. This suggests that the synthetic rollouts generated by reward model are not helpful, possibly due to inaccurate reward model leading to noisy rewards.

### E.5 Varying Selection Fractions

To obtain a better estimate of the trade-offs between training time and performance improvements, we vary the selection fraction in $[\frac{1}{32}, \frac{1}{16}, \frac{1}{8}]$ and present results for two `DQN` configurations with the 125M proxy model: (1) DQN with `Mean-Std` state encodings, `Furthest` subsampling, and an `MLP` (**DQN (F)**), and (2) DQN with `Concat` state encodings, `Furthest` subsampling, and a `Transformer` (**DQN-T (F)**) in Figure 4. For comparison, we also include results for the **Random** and **Full** baselines. The reported wall-clock times account for the combined duration of training the DQN and subsequently training the target model on the selected data subsets, while the wall-clock times for the random baseline include only the target model's training time.

Our results show that with a $\frac{1}{32}$ selection fraction, the DQN-based approaches do not outperform the random baseline and take longer to run. However, for selection fractions $\frac{1}{16}$ and $\frac{1}{8}$, the DQN-based approaches outperform the random baseline, with an additional hour of training time. Although training on the full dataset yields the best performance, it requires more than twice the time needed for the DQN-based approaches with a $\frac{1}{8}$ selection fraction. Notably, while `Transformer`-based DQNs take slightly longer to train, they outperform `MLP`-based DQNs for the $\frac{1}{8}$ selection fraction.

