# OpenReview forum: "RL-Guided Data Selection for Language Model Finetuning"
_NeurIPS.cc/2025/Workshop/Reliable_ML — NeurIPS 2025 - Reliable ML Workshop_

### Official Review · Reviewer_idiE · 2025-09-15
**An RL-based dataset selection for LLM fine‑tuning with strong gains on noisy datasets, while cutting wall-clock training time by up to 2x, showing a viable and promising way towards high-quality dataset selection and efficient training.**

**Rating:** 8
**Confidence:** 4

**Review:**

**Summary**

The paper reframes budget‑constrained fine‑tuning data selection as a tractable MDP by clustering training data, treating the selected cluster set as state, and adding clusters as actions. Rewards are computed via a proxy model’s (MobileLLM‑600M) loss on subsampled train/val data, while policies are learned with DQN, PPO, Dyna‑style model‑based augmentation, and a discrete adaptation of CLIMB. Using MobileLLM‑1.5B as the target model and a 5% budget on four datasets (MMLU, ANLI, MetaHate, GooglePlay), the method matches or beats full‑data training on some tasks while roughly halving wall‑clock time (selection overhead included, HP search excluded). The proposed method achieves an astonishing 10.8 accuracy point gain over full‑data on MetaHate and a slight gain on GooglePlay, while ANLI/MMLU see smaller gains vs. full‑data.

**Strengths**

**Novel, neat formulation and practicality:** The idea of turning subset selection into an MDP over clusters is novel, neat, and computationally tractable at moderate size (64/128), with a reward that is cheap enough to evaluate and reasonably correlated with downstream performance. Details of state encodings, reward variants, and policy learning are well‑described.

**Empirical promise on unreliable data:** On noisy/redundant datasets, RL‑guided selection substantially outperforms random/loss heuristics and sometimes the full dataset. This gives empirical promise on unreliable data, and provides useful insights towards how we quantitatively measure the reliability and quality of data.

**Efficiency:** 5% subsets deliver competitive performance in <50% wall‑clock time vs full‑data training, highlighting a compelling cost–quality trade‑off.

**Insightful:** Though preliminary at the current stage, it can be clearly seen that the paper delivers many interesting insights that are worth further exploration in its full version.

**Weaknesses / Limitations**

**Limited experiments:** Current experiments are only performed on MobileLLM‑600M as proxy and MobileLLM‑1.5B as target (totally understandable as a short research paper, though).

**Mixed consistency across tasks:** ANLI and MMLU lag the full‑data baseline, raising questions about when RL‑guided selection helps or hurts.

**Scalability:** Even with subsampling/log‑transform, repeated proxy retraining per rollout is expensive. The paper proposes model‑based rollouts, but ablation on overall selection cost vs. benefit is limited.

**Suggestions for Authors**

I do like the paper for the novel idea and insights, and see the potential that its full version makes an impact in the AI community if it can address most of the suggestions below.

(1) Dig deeper into the following insights delivered in this paper: (1) quantitative metrics on data quality and reliability; (2) tradeoff between dataset size and reward computation cost in the formulation; (3) design of possible detection and "early-stop" mechanisms when the proposed method doesn't work well (e.g. ANLI, MMLU); (4) what is the amount of data we actually need for a plausible performance, and what is the tradeoff between quantity and quality when quality comes at a cost.

(2) Whether or not the conclusion generalizes to different model families and model sizes is interesting (e.g. Does the reward function help when we use a 600M model as a proxy for a 70B target model?)  More experiments on different model families, different model sizes, different dataset size budget, and different tasks will surely help solidify the paper and offer more value to the industry.

(3) It will be interesting and helpful to do more ablation studies on how we can find out the optimal combination of proxy-target models, how we determine the size of a cluster, how we can determine the optimal dataset size budget for most cost-efficient training, etc.

(4) Is there any optimization we can do on the current RL reward function? Is there any optimization we can do for the reward calculation?

---

### Official Review · Reviewer_CejJ · 2025-09-19
**RL-based method for data selection seems interesting as well as experiment results. I would strongly advise comparing to greedy methods for data selection.**

**Rating:** 6
**Confidence:** 3

**Review:**

Summary: This work tackles the problem of data subset selection by taking the intractable budget-constrained optimization problem and formulating it as an RL problem by defining appropriate states, transitions, and reward function. The method clusters the training dataset samples using K-means clustering, and a state is a subset of clusters selected, and the transition includes a new cluster not already in the current subset. The reward function uses a proxy model M’ to calculate a change in validation loss by including the new cluster to the existing subset (the current state). Experiments are done on four datasets using different RL solvers, with some promising results.

Strengths:
- RL-based method for subset selection appears to be a novel idea, and it’s interesting to see how it can sometimes improve fine-tuning (Google, MetaHate).
- The experiment results are interesting and it’s nice that there is diversity in the datasets chosen and some discussion of possible explanations for the difference gaps between full/RL-chosen subset vs. other heuristics like random or random-search.

Weaknesses:
- There appear to be several external components which influence the effectiveness of this approach, namely the K-means clustering and the proxy model M’ which informs the reward function. These are assumed to be “ideal” but some ablations here would help shed more light on the method.
- The paper would be strengthened by comparison to other methods for subset selection, e.g., there are more sophisticated methods than random, random-search such as certain bi-level optimized methods. There is a vast literature, so it would be good to discuss and compare to some other method(s). In particular, I believe there exist methods which greedily add datapoints also using some proxy of validation loss (which is most similar to this work).